# Current Treatment Methods in Hepatocellular Carcinoma

**DOI:** 10.3390/cancers16234059

**Published:** 2024-12-04

**Authors:** Kamila Krupa, Marta Fudalej, Anna Cencelewicz-Lesikow, Anna Badowska-Kozakiewicz, Aleksandra Czerw, Andrzej Deptała

**Affiliations:** 1Students’ Scientific Organization of Cancer Cell Biology, Department of Oncological Propaedeutics, Medical University of Warsaw, 01-445 Warsaw, Poland; s088253@student.wum.edu.pl; 2Department of Oncological Propaedeutics, Medical University of Warsaw, 01-445 Warsaw, Poland; marta.fudalej@wum.edu.pl (M.F.); anna.badowska-kozakiewicz@wum.edu.pl (A.B.-K.); 3Department of Oncology, National Medical Institute of the Ministry of the Interior and Administration, 02-507 Warsaw, Poland; anna.cencelewicz@cskmswia.gov.pl; 4Department of Health Economics and Medical Law, Medical University of Warsaw, 01-445 Warsaw, Poland; aleksandra.czerw@wum.edu.pl; 5Department of Economic and System Analyses, National Institute of Public Health NIH-National Research Institute, 00-791 Warsaw, Poland

**Keywords:** hepatocellular carcinoma, TACE, TARE, radiation therapy, sorafenib, lenvatinib, immunotherapy, pembrolizumab, atezolizumab, nivolumab

## Abstract

Hepatocellular carcinoma (HHC) remains a severe threat to world health due to its delayed detection, complicated treatment, and rapid, asymptomatic progression. The high recurrence rates for surgical resections compel researchers to investigate more effective treatment methods. Clinical trials showed promising results in targeted therapy, minimally invasive procedures, and immunotherapy, leading to an increase in overall survival and progression-free survival in HCC patients. For those who do not qualify for surgery, minimally invasive treatments like transarterial therapies and local ablative therapies provide possibilities. Systemic therapies, including targeted therapies and immunotherapy, are essential for advanced HCC. Moreover, the evaluation of combination therapy is a major point for recent clinical trials. An overview of the current approaches to treating HCC is provided in this review.

## 1. Introduction

Hepatocellular carcinoma (HCC) is one of the most common malignant tumours in the world and the third most common cause of cancer-related deaths, according to Global Cancer Incidence, Mortality and Prevalence (GLOBOCAN) 2020 [1]. HCC occurs in 80–90% of patients with cirrhosis. Significant risk factors include hepatitis B (HBV) and hepatitis C virus (HCV) infection, alcohol liver disease, non-alcoholic fatty liver disease (NAFLD), and non-alcoholic steatohepatitis (NASH) [2]. Other significant contributors are aflatoxin exposure, specific metabolic conditions, and type 2 diabetes [3]. HCC is asymptomatic in its early stages, which leads to late diagnosis [1]. The triangular presentation of right upper quadrant pain, palpable mass, and weight loss characterizes 90–95% of HCC patients. The other physical symptoms include hepatic bruits (25% of patients), ascites, splenomegaly, jaundice, fatigue, and fever. The functional reserve capability of the liver means that jaundice and liver function tests might not show up for a long time in the progression of the disease [4]. Numerous treatment methods for HCC are recommended, depending on the stage of the tumour and liver function. The performance status and liver function of patients with HCC have a significant role in their prognosis. Thus, the modified Barcelona-Clinic-Liver Cancer (BCLC) Classification method is used to categorise HCC in liver cirrhosis [5]. Conventional chemotherapy and radiation therapy are notably ineffective against HCC. The U.S. Food and Drug Administration (FDA) has approved transarterial chemoembolization (TACE), radiofrequency ablation (RFA), and surgical resection as therapies for advanced HCC. In contrast, liver transplantation is the preferred course of action in the preliminary stages of the disease. Tyrosine kinase inhibitors (TKIs) like lenvatinib and sorafenib, in combination with immunotherapy and anti-angiogenesis therapy like bevacizumab and atezolizumab, are additional first-line treatments for advanced HCC [1]. This review elaborates on the current status of HCC treatment, focusing on the latest clinical trials and advanced techniques. The most common methods, which are described in this review, are presented in Figure 1.

## 2. Surgical Resection and Liver Transplantation

Advanced surgical techniques and instrumentation have improved the surgical treatment of HCC; however, the best candidates for resection are patients with very early or early-stage HCC. Major liver resection is possible, provided the organ is in good functional condition and has adequate liver reserves. Careful evaluation of the liver for signs and symptoms related to portal hypertension is necessary, which is crucial in determining the relationship of the tumour to the main vasculature of the liver [6]. To avoid liver failure, it is recommended to maintain a minimum residual liver volume, which is 40% for cirrhotic patients and 20% for non-cirrhotic patients. Resection is associated with a high recurrence rate, which ranges between 40% and 80% within 5 years of surgical resection. For patients with early recurrence, palliative therapies are used, while patients with late recurrence may benefit from re-resection [7].

Patients with cirrhosis are referred for liver transplantation (LT), provided they meet the Milan criteria (MiC). The liver should have one tumour smaller than 5 cm or three tumours smaller than 3 cm each, without vascular invasion or extrahepatic spread [8]. Liver transplantation gives a better oncological result than surgical resection because it not only removes all pre-cancerous and cancerous lesions in the liver but also treats coexisting liver disease [7]. Compliance with MiC criteria significantly increases the 5-year overall response rate (ORR) to 68% and reduces the relapse rate to less than 15% [7]. The disadvantages of LT include the waiting time for the liver, during which HCC progression may occur, including vascular invasion and with further failure to meet the criteria. Bridging therapies such as RFA or TACE are used to slow cancer progression when the estimated waiting time for a transplant is longer than six months [6,8]. Potential solutions to the problem also include increasing the donor pool through living donation. Living donor liver transplantation (LDLT) has now become an alternative to deceased donor liver transplantation (DDLT); nevertheless, the risk of donor mortality or possible problems with matching the size of the graft to the recipient area must be carefully assessed [6,8].

## 3. Minimally Invasive Procedures

In patients with HCC and cirrhosis, who are not qualified for major surgical procedures, minimally invasive treatment procedures can be offered, because of their efficiency [9]. According to the BCLC staging system, minimally invasive procedures have been indicated for stages 0, A, and B [10] (Table 1).

### 3.1. Local Ablative Therapies

Local ablative therapy is classified into two categories: thermal ablation and chemical ablation [6]. These techniques include RFA, cryoablation, microwave ablation (MWA), high-intensity focused ultrasound (HIFU), percutaneous ethanol injection (PEI), percutaneous acetic acid injection, irreversible electroporation (IRE) and laser-induced thermotherapy [11]. RFA and MWA are the most common forms of thermal ablation [12]. Ablation is indicated on patients with small HCC, up to three tumours each ≤3 cm in diameter, with Child-Pugh class A or B [13].

PEI is a well-tolerated, low-cost treatment. It involves using computed tomography (CT) or ultrasound imaging to guide a needle into a tumour. When administered, concentrated ethanol directly damages tumour cells by denaturing and dehydrating them, resulting in coagulative necrosis [14]. These days, ethanol injection is the preferred course of treatment only when RFA is not possible due to enterobiliary reflux, tumour adherence to the gastrointestinal tract, or various other reasons [15]. High-frequency alternating current is used in RFA to eliminate solid tumour tissue. Heat is produced from the radiofrequency energy released by the electrode’s exposed tip. Heat conducts equally in all directions. Electrodes come in three forms: perfusion, internally cooled, and multitined expandable. Lately, RFA has been the most widely used ablation method for HCC [13,15]. Currently, it is evaluated in combination with TACE. In El Dorry et al. (2021)’s study, patients with stage B HCC were treated with RFA + TACE, which, in comparison to TACE alone, showed an effective complete response, overall survival (OS), downstaging, and disease-free survival (DFS) [16]. Several studies and meta-analyses confirm the improvement in treatment with RFA + TACE combination also in comparison to RFA monotherapy [17,18,19].

Every method of destroying tumours that requires technology to produce electromagnetic radiation (EMR) at frequencies more than 900 kHz is collectively referred to as MWA. Instead of damaging DNA as ionizing radiation does in stereotactic body radiation therapy (SBRT), this lower-energy radiation heats the tissue it passes through. When one gets farther away from the source, the amount of heat damage decreases rapidly. This is categorised as thermal ablation in the same way as RFA. Crucially, arteries close to the tumour are less vulnerable to damage from MWA than with RFA [14]. The similar diameters of ablative regions, similar recurrence-free survival (RFS), and similar OS at 50 months in liver tumours between 1.5 and 4 cm were reported in a phase II clinical trial [20]. According to Xiong et al. (2024)’s study, RFA can offer a better prognosis than MWA for patients with cirrhotic HCC, with a reduced mortality and recurrence rate [21]. Although the RFA and MWA are mostly similar, further studies should be conducted in patients with larger tumours.

For unresectable HCC, the HIFU technique has shown efficacy [22]. The mechanism of action is to deliver mechanical energetic waves to targeted HCC, which causes parenchymal necrosis induced by heat shock due to local heat waves. The limitation of this method is the small size of the focus volume, which demands multiple treatment sessions [9]. However, compared with other ablation therapies, it is the only noninvasive technique for the treatment of solid tumours. One of the studies compared the combination of HIFU and TACE with surgical resection for a single HCC with Child-Pugh B cirrhosis. The results showed that the surgical resection still has a definitive advantage of delaying recurrence in comparison to TACE combined with HIFU. It may be associated with the residual tumour following the procedure. On the other hand, the combination had higher safety than surgical resection [23].

### 3.2. Transarterial Therapies

The liver without lesions receives 75% of its blood supply from the portal vein and the rest 25% from the hepatic artery. On the other hand, the HCC derives most of its blood supply (about 90%) from the hepatic artery, so the obstruction of this artery could induce tumour necrosis due to a lack of blood [6,24]. Transarterial therapies like TACE, transarterial embolization (TAE), and transarterial radioembolization (TARE) are widely used treatment modalities for HCC. According to the BCLC, TACE is indicated in intermediate-stage HCC (stage B) with nodules smaller than 3 cm in diameter without vascular involvement. It is known as a bridging therapy for HCC patients awaiting liver transplantation to maintain the eligibility criteria [9,24]. This technique is contraindicated in patients with bilirubin levels more than 3 mg/dL, advanced cirrhosis, complete portal vein thrombosis, liver failure, and the presence of extrahepatic disease like renal insufficiency and severe atherosclerosis because of the possibility of hepatic damage and liver decompensation [25,26]. TACE techniques encompass conventional TACE (cTACE) and drug-eluting beads, which use TACE (DEB-TACE). The cTACE is considered a standard treatment with the highest grade of recommendation, and it is based on the intra-arterial injection of cytotoxic agents in the lipiodol [27,28]. The DEB-TACE includes the novel drug delivery embolization system, which provides a more sustained release of the drug into the tumour and a lower release into the systemic circulation [29]. The comparison of the efficacy of cTACE and DEB-TACE in the PRECISION V phase II study, PRECISION ITALIA STUDY GROUP phase III trial, and meta-analysis from four randomised controlled trials and eight observational studies showed no significant difference between the two treatments in terms of tumour response and survival rates [28,29,30,31]. The innovative balloon-occluded TACE (B-TACE) technique involves using the microballoon catheter to infuse the chemotherapeutic emulsion with lipiodol followed by gelatine corpuscles under the occluded artery. It is a promising treatment for HCC patients with only a few nodules [32]. Some studies investigated the combination treatment with TACE. The results from a meta-analysis of eight trials showed that the combination of TACE and RFA was more effective than RFA alone in HCC patients with intermediate and large-size tumours and patients at younger ages [33]. Based on the results from a meta-analysis of 25 studies, in patients with unresectable HCC, TACE plus radiation therapy (RT) should be recommended [34]. Moreover, the meta-analysis compared the combination of TACE plus MWA with TACE monotherapy for BCLC stage A or B HCC patients with nodules lesser than 5 cm. The results showed significant improvements in complete response, partial response, and ORR for the TACE plus MWA treatment [35]. Most trials are investigating the combination of TACE with anti-angiogenic agents, which may delay the progression of the tumour. The results from the TACTICS trial (NCT01217034), including patients with unresectable HCC, showed that the median progression-free survival (PFS) was significantly longer in the TACE plus sorafenib than in the TACE alone group (25.2 vs. 13.5 months; *p* = 0.006) [36].

TARE refers to the injection of radioactive substances through the hepatic artery: microspheres containing yttrium-90 (Y-90) or iodine-131 and iodised oil [27,37]. The procedure can achieve different degrees of regression in 25–50% of HCC patients [27]. TARE is known as a safe and effective treatment for unresectable HCC, as it has a safer toxicity profile than TACE, longer time to progression (TTP), and greater ability to bridge therapy for HCC patients awaiting LT [37,38]. However, the two treatments do not significantly differ in terms of OS [39]. Phan et al. (2024) compared TACE and TARE as first-line treatments for unresectable HCC greater than 8 cm. Although the ORR and disease control rate (DCR) were similar in both groups, the safety profile differed significantly. Major AEs occurred in the TACE group (72% vs. 5%; *p* < 0.001), including post-embolization syndrome (100% vs. 75%, *p* = 0.002). The results suggest that TARE provides a safer profile than TACE [40]. TARE was also compared with DEB-TACE in the phase II TRACE trial (NCT01381211) (in a group of patients with intermediate-stage or early-stage HCC patients. The median TTP and median OS were significantly longer in the TARE group (17.1 vs. 9.5 months; 30.2 vs. 15.6 months). Serious AEs occurred in 39% of TARE patients versus 53% of DEB-TACE patients [41]. TARE can also be chosen as an alternative to ablation and chemotherapy [37]. The RASER study (NCT03248375) showed that ORR and complete response rate were 100% and 90%, respectively, in the group of patients with unresectable early HCC, who were not candidates for RFA. The OS (1-year and 2-year) was 96%, and the AEs occurred in 7% of patients [25]. Y-90 RE can be also used as a neoadjuvant treatment for stage C HCC patients with portal vein tumour thrombosis (PVTT). Martelletti et al. (2021) compared TARE with sorafenib in patients with HCC and intrahepatic PVTT and found that TARE was more effective in downstaging patients to surgery and improved OS [42]. Spreafico et al. (2018) found that the combination of bilirubin levels, an extension of PVTT, and tumour burden might be a guide to identifying the best candidates for the treatment [43]. The latest clinical trials compared the effectiveness of the combination therapy. In the phase II SORAMIC (NCT01126645) trial, the treatment with sorafenib with RE resulted in a higher ORR (61.6% vs. 29.8%; *p*  <  0.001), complete response rate (13.7% vs. 3.8%; *p*  =  0.022), longer PFS (8.9 vs. 5.4 months; *p*  =  0.022), hepatic PFS, and TPP in comparison to sorafenib monotherapy. However, the results did not translate into prolonged OS, and the Child–Pugh B patients had lower response rates [44].

### 3.3. Radiation Therapy

RT has evolved over the past few decades due to improved imaging and advancements in technology so that it can provide high local control rates in unresectable HCC, palliative therapy in metastatic cases, and can help bridge patients to LT or curative resection. Currently, external beam radiation therapy (EBRT), conventional EBRT radiation techniques like intensity-modulated radiotherapy (IMRT) and three-dimensional conformal radiotherapy (3DCRT), SBRT, proton beam therapy (PBT) and selective internal radiation therapy (SIRT) are used in HCC treatment [45]. Currently, radiotherapy is being compared to well-known methods, and its effectiveness in combination with other HCC therapies is being investigated.

According to the Sapir et al. (2018) study, SBRT may offer higher local control rates, so it can be an alternative treatment for TACE in patients with BCLC stage B HCC [45,46]. The phase II SBRTvsTACE study (NCT02182687) compared SBRT to TACE as a bridging strategy for patients with HCC undergoing orthotopic liver transplantation. Combination therapies involving radiation therapy have shown promising results. Several reports suggest that the combination of TACE and SBRT might yield better results than TACE alone, but it could also increase the risk of treatment-related adverse events [34,46,47]. According to the systematic review and meta-analysis, tumour size and region were associated with the OS. The analysis indicates that SBRT is effective for smaller tumours <3 cm, Eastern region, Child–Pugh score ≤ B7, and BCLC stage 0 or A [48]. Additionally, the phase 3 RTOG 1112 trial (NCT01730937) assessed the usage of SBRT followed by sorafenib versus sorafenib monotherapy in patients who are not candidates for RFA and TACE. SBRT improved the OS (12.3 vs. 15.8 months), PFS (5.49 vs. 9.22 months), and quality of life at 6 months, with no increase in AEs [49].

EBRT can offer good local control, especially in unresectable HCC [45]. Shirono et al. (2021) discussed the safety and efficacy of a combination of hepatic arterial infusion chemotherapy (HAIC) and EBRT for unresectable HCC involving the inferior vena cava (IVC) and right atrium (RA). The results showed an efficacious treatment option for unresectable HCC, even complicated with pulmonary embolism. Therefore, after the examined treatment, immuno-oncology therapy should be applied [50]. In the Li et al. (2019) study, the retrospectively reviewed HCC patients with IVC/RA tumour thrombosis were treated with EBRT, where the size was smaller than 10 cm. In comparison to surgery, radiotherapy may predict a longer TPP [51]. Su et al. (2023) compared the effectiveness of EBRT versus TACE for HCC with a tumour diameter ≥ 5 cm, highlighting the importance of selecting the appropriate treatment modality based on tumour characteristics. EBRT manifests to be more successful than TACE as a primary local treatment for HCC with a tumour diameter between 5 and 10 cm [52].

Proton beam therapy (PBT) and magnetic resonance image-guided radiation therapy (MRgRT) have enhanced the capabilities of EBRT. MRgRT is useful in reducing high-dose radiation to normal tissue and provides soft tissue delineation in comparison to CT. PBT, by using a beam of proton particles, protects the uninvolved liver by offering to minimise low-/moderate-dose radiation and dose escalation. That may result in better survival in HCC patients compared to photon therapy, which uses high-energy X-rays to send the radiation inside the body to the tumour. The results of the phase III NRG Oncology GI-003 trial (NCT03186898) will show the differences in treating HCC patients with IMRT versus PBT [53].

In the case of lesions more than 7 cm in diameter, with vascular invasion or failure of prior TACE, SIRT is used for HCC patients. The procedure comprises an injection of microspheres labelled with a b-emitting radioisotope, such as Yttrium-90, which delivers the radiation to the tumour [9,45]. Several studies have been conducted to evaluate the efficacy and safety of SIRT in patients with various stages of HCC. In the phase III SIRveNIB study (NCT01135056), the efficacy of SIRT versus sorafenib was compared in patients with locally advanced HCC. The OS did not differ significantly between the groups, although the toxicity profile was improved in the SIRT group [54,55]. The purpose of the SARAH study (NCT01482442) was to compare the safety and efficacy of sorafenib to SIRT with Yttrium-90 in patients with BCLC stage C HCC or new HCC not eligible for surgical resection, LT, or thermal ablation after a previous treatment involving surgery or thermoablative therapy, or HCC with two unsuccessful rounds of TACE. The results showed that OS did not significantly differ between the two groups, so tolerance might help in choosing the proper treatment [56].

## 4. Systemic Therapy

The treatment is relevant for advanced-stage HCC or when cancer cannot be treated with localised therapies like surgery or locoregional therapy. The main types of systemic therapies include targeted therapies and immunotherapy. Child–Turcotte–Pugh A patients with good performance status have been included to assess the effectiveness of these treatments; however, well-selected patients with Child–Turcotte–Pugh B7 are capable of tolerating those therapies [10]. Chemotherapy with doxorubicin, gemcitabine, or combined is less commonly used for HCC due to low effectiveness, significant side effects, and acquired drug resistance; however, it still may be an option for HCC patients who progressed on sorafenib treatment [6,57].

### 4.1. Targeted Therapies

Sorafenib is an oral multikinase inhibitor of the serine-threonine kinases Raf-1 and B-Raf, as well as the receptor TKI of vascular endothelial growth factor receptors (VEGFRs)-1, 2, and 3, and platelet-derived growth factor receptor beta (PDGFR-beta). The kinases play a crucial role in cellular signalling linked to the molecular pathogenesis of HCC [58,59]. Results from the multicenter, phase III, double-blind, placebo-controlled SHARP trial (NCT00105443) led to FDA approval in 2007 for therapy with sorafenib for advanced HCC as the first-line standard-of-care treatment (Table 2) [58,60]. In the GIDEON study, sorafenib was evaluated in Child–Pugh A and Child–Pugh B patients. The safety profile indicated that patients with Child–Pugh B may safely receive the treatment [61]. Since then, several analysed TKIs, including brivanib, sunitinib, and linifanib, have not demonstrated equivalence with sorafenib. The phase I/randomised phase II trial (NCT01004003) evaluated maximum tolerated dose (MTD) per dose-limiting toxicities (DLT-s) as well as the safety and efficacy of nintedanib (BIBF 1120) vs. sorafenib in European patients with unresectable advanced HCC. Although the tolerability, dose intensity, and incidence of drug-related AEs and grade ≥ 3 AEs favoured nintedanib, the number of AEs leading to discontinuation was greater in this group of patients. Moreover, nintedanib was related to higher VEGF-related toxicity [62]. Promising outcomes from studies on lenvatinib, an inhibitor of VEGF receptors 1–3, fibroblast growth factor receptors (FGFRs) 1–4, PDGFR-alpha, RET, and KIT, led to the approval of the first alternative for sorafenib by the FDA in August 2018. In the phase III, open-label, multicenter, non-inferiority REFLECT clinical trial (NCT01761266), OS was evaluated as the primary goal between patients with unresectable HCC who received lenvatinib and those who received sorafenib. Comparing lenvatinib to sorafenib, the median survival time was not worse. Furthermore, comparable occurrences and types of adverse events were documented, apart from proteinuria and grade 3 or 4 hypertension in the lenvatinib group (Table 2) [58,63].

The efficacy of lenvatinib treatment is under investigation. In the analysis of 313 patients with advanced HCC treated with lenvatinib between 2019 and 2022, the OS and MPF were higher in the nonviral aetiology group. The OS was 21 months in the group with other aetiologies versus 15 months in the group with viral aetiology, and the MPF was 9 months and 6 months, respectively. Moreover, the toxicity profile did not change. Lenvatinib therapy showed greater effectiveness in the case of nonviral aetiology, which may have an impact on the selection of appropriate therapy [64]. The retrospective analysis of patients with non-viral advanced HCC treated with atezolizumab plus bevacizumab, lenvatinib, or sorafenib demonstrated that OS and PFS were significantly longer in the group of patients treated with lenvatinib versus atezolizumab plus bevacizumab. After dividing the cohort into two groups, NAFLD/NASH and non-NAFLD/NASH, the analysis confirmed that the longer OS and PFS were corelated with lenvatinib treatment in the NAFLS/NASH population compared to atezolizumab plus bevacizumab. In the non-NAFLD/NASH group, as in the comparison of patients receiving atezolizumab plus bevacizumab versus sorafenib, no statistically significant differences were observed [65]. Another study compared the efficacy of lenvatinib to the combination of atezolizumab plus bevacizumab in a large cohort of patients with Child Pugh B class HCC. The median OS was greater in the group of patients receiving Lenvatinib, 13.8 months, than in the group receiving atezolizumab plus bevacizumab, 8.2 months [66].

For the second-line therapy, after progression with sorafenib, the systemic therapeutic options include regorafenib and cabozantinib. The first one in the RESORCE phase III trial (NCT01774344) has shown improvement in OS in comparison to placebo in the group of patients with HCC who had been previously treated with sorafenib, with minimal stabilization of 7 months [67]. Cabozantinib targets VEGF, AXL, and MET receptors. The AXL and MET receptors have been implicated in antiangiogenic resistance and they are correlated with poor prognosis. In the CELESTIAL phase III clinical trial (NCT01908426), it also improved efficacy outcomes, the OS, PFS, and the objective response rate (ORR) versus placebo in the group of patients with HCC who had been previously treated with sorafenib. However, the rate of high-grade adverse events was observed in the cabozantinib group. The OS was 10.2 versus 8.0 months, PFS 5.2 vs. 1.9 months, respectively. In comparison to the RESORCE trial, where serious AEs occurred in 51.9% patients treated with regorafenib versus 47.7% in the placebo group, in the CELESTIAL trial, serious AEs appeared in 49.7% patients treated with cabozantinib versus 36.7% in the placebo group. The most common grade 3 or 4 AEs in both trials were hypertension, palmar-plantar erythrodysesthesia, fatigue, and diarrhoea. Given the circumstances, cabozantinib has more side effects than placebo but not more than regorafenib [68,69].

### 4.2. Combination Therapies

Combination therapy is also being investigated. In the randomized, double-blind, placebo-controlled phase II study (NCT01258608), the combination of sorafenib and mapatumumab was evaluated in patients with advanced HCC. Due to a lack of improvement in TPP, other efficacy end points, like OS, PFS, or change in toxicity profile, further investigation of this combination is not arranged [70]. The phase III NRG/RTOG 1112 trial (NCT01730937) assessed sorafenib combined with SBRT compared to sorafenib alone. The results showed efficacy by improving the OS and PFS in the group treated with the combination therapy. Moreover, an increase in adverse events (AEs) was not observed [49]. In the phase III LAUNCH trial (NCT03905967), the combination of TACE + Lenvatinib resulted in significantly longer median OS in comparison to the lenvatinib alone group [71]. More studies based on the combination of sorafenib with other therapies should be conducted.

### 4.3. Sorafenib and Ferroptosis

Ferroptosis, a recently identified form of cell death, is driven by a large amount of iron accumulation and lipid peroxidation, and it has been recently corelated with the tumour resistance mechanism. Glutathione peroxidase (GPX), especially GPX4, neutralises reactive oxygen species (ROS) and reduces lipid peroxides. Its downregulation increases the sensitivity to ferroptosis due to the accumulation of lipid peroxides [72]. Sorafenib indirectly promotes ferroptosis by inhibiting the solute carrier family 7 member 11 (SLC7A11) transporter, which is essential for cystine uptake on the cell membrane. Reduced amounts of cystine in HCC cells limit glutathione (GSH) synthesis, leading to diminished activity of GPX4 and ferroptosis [73]. HCC patients treated with sorafenib often develop resistance, limiting the long-term effectiveness of the drug. It is related to such transcription factors and genes as nuclear factor erythroid 2-related factor 2 (Nrf2), Rb, metallothionein 1-G (MT-1G), and sigma 1 receptor (S1R), which regulate the sensitivity of cells to ferroptosis, so novel inhibitors against these factors may overcome the drug resistance to sorafenib [73,74].

## 5. Immunotherapy

As previously mentioned, resection or transplantation has limited effectiveness due to a low classification rate for surgery and a high 5-year recurrence rate, ranging up to 70% after surgery [75]. It is related to the late diagnosis of HCC, which, in most patients, is at an advanced stage. Treatment is then based on a combination of targeted therapy, chemotherapy, or radiotherapy. Recently, immunotherapy has also been included in the treatment of HCC (Table 3).

Continuous exposure of the liver to bacterial components and dietary antigens from the gastrointestinal tract led to the creation of an immune microenvironment consisting of Kupffer cells, hepatic stellate cells, sinusoidal hepatic endothelial cells, natural killer (NK) cells, gamma-delta T cells, and dendritic cells [76]. Cytokines, growth factors, chemokines, and interactions with the liver’s immunological environment trigger the defensive response during infection or injury; if these processes are not properly regulated, cancer may develop [77]. Zhang et al. in their study found that hypoxia- and inflammation-related hypoxia-inducible factor 1α (HIF-1α) can stimulate the excessive expression of IL-1β in tumour-associated macrophages (TAMs), which, thanks to positive feedback, can induce the production of HIF-1α and facilitate the process of epithelial–mesenchymal transition (EMT), leading to metastasis [78]. Other important cytokines involved in the pathogenesis of HCC are IL-6, IL-11, and lncRNA activated by TGF-β (lncRNA-ATB) [77]. Patients with HCC usually have an increased number of Tregs, which can inhibit the function of CD8 + T cells and further elimination of tumour cells [79]. Additionally, continuous antigen stimulation leads to the exhaustion of T cells and, thus, an increase in the expression of co-inhibitory signalling molecules, such as cytotoxic T-lymphocyte-associated protein 4 (CTLA-4), programmed cell death protein 1 (PD-1), and lymphocyte activation gene 3 (LAG-3) [80].

The study based on clinical trials of immune checkpoint inhibitors, i.e., monoclonal antibodies directed against extracellular ligands involved in suppressing the antitumour immune response, allowed for the recognition of three categories of molecular targets—PD-1, CTLA-4, and LAG-3—which are intended to restore the antitumour capacity of lymphocytes T [81].

Nivolumab is a human anti-PD-1 IgG4 monoclonal antibody directed against PD-1 [82]. After approval by the FDA in 2017, it is used as a second-line therapy in the treatment of HCC. The CheckMate 040 (NCT01658878) study evaluated nivolumab in combination with ipilimumab in patients treated with sorafenib as a first-line therapy in the treatment of advanced HCC, while the CheckMate 459 study (NCT02576509) compared nivolumab with sorafenib. The results from the first study led to FDA approval in 2020 of this regimen for the treatment of patients with advanced HCC who were previously treated with sorafenib [83]. Nivolumab showed, comparable to Child–Pugh A, a safety profile and clinical activity in patients with Child–Pugh B advanced HCC [84]. Furthermore, nivolumab monotherapy continued to provide a durable therapeutic benefit in patients with advanced HCC, with or without prior sorafenib treatment with 5 years of follow-up [85]. The results of the second study showed no significant difference in OS; nevertheless, the use of immunotherapy showed a lower incidence of grade 3 to 4 adverse effects and improvement in patients’ quality of life; thus, nivolumab may be an alternative treatment option for patients who cannot undergo therapy with tyrosine kinase inhibitors or antiangiogenic therapy (Table 4) [86]. The monoclonal antibody atezolizumab binds to the PD-L1 protein (programmed death-ligand 1), blocking the binding site of PD-1. The IMbrave150 study (NCT03434379) compared atezolizumab therapy in combination with the antiangiogenic bevacizumab with a group of patients with locally advanced or metastatic HCC treated with sorafenib. This study showed statistically significantly better OS and PFS with monoclonal antibody immunotherapy compared to sorafenib (Table 4) [87,88]; however, Child–Pugh B patients were not included in the study. In 2020, the combination of atezolizumab plus bevacizumab became a new standard of care for first-line unresectable HCC patients with Child–Pugh A cirrhosis. The benefits of the therapy were assessed in a meta-analysis in subsequent years [89]. Data from the IMbrave150 study, along with real-world data from patients who underwent tumour resection after treatment, were used in the study to evaluate long-term outcomes in HCC patients who had stable disease (SD) or durable partial response (PR) following therapy with atezolizumab plus bevacizumab. The results suggest a favourable outcome for patients with durable PR in the IMbrave150 study and in the real-world setting, where half of the persistent PR tumours seemed to be ghost tumours, radiologically persistent tumours lacking viable tumour cells. Ghost tumours may still be linked to patients with persistent SD (16.7%). The results of the study indicate that there is a significant discrepancy between radiological and pathological responses to immunotherapy, suggesting that current imaging methods are insufficient to detect pathologic complete response (PCR). The identification of ghost tumours following immunotherapy would significantly help clinical decision making regarding subsequent treatment modalities. Ghost tumour detection appears to be effective with innovative imaging methods and biomarkers [90]. The problem with high rates of recurrence after ablation or resection is a new challenge for HCC treatment. The phase III IMbrave050 trial demonstrated an efficacious adjuvant therapy using atezolizumab plus bevacizumab for HCC patients who have undergone surgical resection or ablation. When compared to patients in the active surveillance arm, the chance of mortality or recurrence was decreased by 28%. Moreover, the 12-month RFSs were 78% in the atezolizumab plus bevacizumab group versus 65% in the active surveillance group, but a longer follow-up is necessary to verify if the RFS will be maintained [91].

Another known monoclonal antibody is pembrolizumab. The phase II KEYNOTE-224 study (NCT02702414) tested the effectiveness of pembrolizumab monotherapy in patients with HCC previously treated with sorafenib (cohort 1) and with no prior systemic therapy (cohort 2), and the results in the study groups were better in terms of safety profile and OS [92,93]. Similar effectiveness and safety were demonstrated by pembrolizumab in combination with the best supportive care (BCS) in the phase III KEYNOTE-394 study (NCT03062358), where the study group included Asian patients previously treated with sorafenib or oxaliplatin-based chemotherapy. Compared to the placebo plus BCS group, pembrolizumab showed a statistically significant improvement in OS and PFS, as well as in the ORR (Table 4). The results of these trials showed that pembrolizumab monotherapy consistently improved clinical outcomes in patients with advanced HCC. The phase III KEYNOTE-937 study (NCT03867084), where pembrolizumab is being tested as an adjuvant therapy, is under way [94].

Tislelizumab, one of the PD-1 inhibitors, has shown durable clinical activity in the phase II RATIONALE-208 study (NCT03419897) in advanced HCC patients who had undergone prior systemic therapy. The positive outcome led to profound investigation of ICI for first-line monotherapy [95,96]. The phase III RATIONALE-301 trial (NCT03412773) compared tislelizumab versus sorafenib as a first-line treatment for unresectable HCC. Although the ORR and median duration of response were higher in the tislelizumab group, the median PFS was longer in the sorafenib group. The superiority of OS for tislelizumab was not met; however, the safety profile of tislelizumab was more favourable than that of sorafenib [97].

The CTLA-4 protein, a CD28 homolog, prevents the binding of CD28 to CD80 and CD86, which is necessary for optimal T-cell activation. Moreover, it reduces the activity of helper T cells, increasing Treg activity, which leads to suppression of the immune response. A phase I/II study (NCT02519348) compared tremelimumab, a monoclonal antibody against CTLA-4, in combination with durvalumab (T300 + D) with tremelimumab or durvalumab monotherapy. The results showed that combination therapy was associated with overall treatment benefits and the most encouraging benefit–risk profile due to unique pharmacodynamic activity [98]. Durvalumab monotherapy and the T300 + D regimen were compared to sorafenib in the phase III HIMALAYA trial (NCT03298451). T300 + D, for which a single dose, known as STRIDE (Single Tremelimumab Regular Interval Durvalumab), displayed efficacy and a favourable benefit–risk profile versus sorafenib (Table 4) [99]. The regimen of two ICIs that block CTLA-4 and PD-L1 is the first-line treatment for adults with advanced or unresectable HCC in the EU [100].

LAG-3, present in CD8 + T cells, binds to MHC class II molecules, and its increased expression correlates with the dysfunction of T cells [77]. Zhou et al. showed that after using antibodies against PD-L1, TIM-3 (mucin domain containing-3), or LAG-3, T lymphocytes respond to HCC antigens [101]. TIM-3 is present in TAM cells, and its increased expression correlates with poor prognosis for HCC patients [102]. The effectiveness of TIM-3-targeted therapies remains uncertain, but studies are currently underway using the anti-TIM-3 antibody, colbolimab, LY3321367, or sabatolimab [103].

Due to the role of TGF-β in creating a tumour microenvironment conducive to growth and metastasis, as well as by hindering the infiltration of T lymphocytes into the tumour centre, blocking the action of this cytokine has become another therapeutic target [77]. The discovery of the synergism of the effect of the TGF-β blockade and anti-PD-L1 antibodies allowed for consideration in a clinical trial (NCT02423343) of the combination of galunisertib with nivolumab in the treatment of recurrent or refractory non-small cell lung cancer (NSCLC) or hepatocellular carcinoma (HCC). The results showed that the combination is well tolerated [104]. The phase III CheckMate 9DW study (NCT04039607) included patients with previously untreated HCC, not eligible for curative surgical or locoregional therapies. The firsts results, after a median follow-up of 35.2 months, demonstrate that the combination of nivolumab plus ipilimumab is a potential new first-line standard of care due to statistically significant OS benefit in comparison to lenvatinib or sorafenib (23.7 months vs. 20.6 months; HR 0.79; 95% CI 0.65–0.96; *p* = 0.0180). Moreover, the ORR was also higher, 36% versus 13%, respectively [105]. The ongoing phase III CheckMate 9DX study (NCT03383458) is investigating if nivolumab will improve recurrence-free survival (RFS) compared to placebo in HCC patients (Child–Pugh score 5 or 6) who have undergone resection or local ablation but are at high risk of recurrence [106].

ICIs can be combined not only with each other but also with radiotherapy, FGFR inhibitors, TKI, SBRT, TACE, or chemotherapy. The combination of ICI + TKI in the first-line setting of metastatic disease was investigated in several phase III trials, such as the LEAP-002 trial (NCT03713593) evaluating lenvatinib plus pembrolizumab compared with placebo in patients with advanced HCC (Table 4) and the phase III CARES-310 trial (NCT03764293) with the evaluation of camrelizumab (SHR-1210) plus apatinib (rivoceranib) versus sorafenib [77,107]. The results of the CARES-310 trial showed significant benefit in OS and PFS in the camrelizumab plus rivoceranib group for patients with unresectable HCC with no prior systemic therapy (Table 4) [108]. Furthermore, in patients with advanced HCC with no prior systemic therapy, the phase III COSMIC-312 trial (NCT03755791) assessed the effectiveness of cabozantinib plus atezolizumab in comparison to sorafenib. Median follow-up was 22.1 months, and although the combination did not improve the OS, the PFS was maintained (Table 4) [109]. The ORIENT-32 trial (NCT03794440) evaluated the combination of sintilimab (a PD-1 inhibitor) plus IBI305, a bevacizumab biosimilar, versus sorafenib. The results showed that the combination has significantly greater OS and PFS than sorafenib, which could provide a novel treatment option for patients with unresectable, HBV-associated HCC [110].

ICI can also be used in combination with TACE, whose safety and effectiveness are being investigated in the phase II/III TACE-3 study (NCT04268888), phase II IMMUTACE study (NCT03572582), phase II TRIPLET study (NCT04191889), phase III TALENTACE study (NCT047126430), phase III LEN-TAC study (NCT05738616), phase III CheckMate 74W study (NCT04340193), phase IIIb ABC-HCC study (NCT04803994), and phase III LEAP-012 study (NCT04246177) [81,111,112]. The last one, the phase III LEAP-012 trial, showed that the treatment with lenvatinib plus pembrolizumab in combination with TACE compared with TACE alone has a clinical benefit in patients with intermediate-stage HCC not amenable to curative treatment. At the data cut-off (30 January 2024), the PFS was significantly improved in the pembrolizumab plus lenvatinib group (HR 0.66; 95% CI 0.51–0.84; *p* = 0.0002), and the median PFS was 14.6 months (pembrolizumab plus lenvatinib) versus 10 months (placebo). The evaluation criteria for the OS have not been reached yet [113,114]. For intermediate-stage HCC, the combination of nivolumab plus TACE is being explored in the phase II single-arm IMMUTACE study. The trial achieved its primary endpoint with an ORR of 71.4%. The mPFS was 7.2 months, and the median time to failure of strategy (mTTFS) was 11.2 months, showing prolonged disease control. The median OS was 28.3 months, so there is an improvement in comparison to the typical TACE treatment. Moreover, the time to subsequent systemic therapy was 24.9 months, which delays requirement for further treatments. The further analysis of the expression signatures, changes in immune cell populations and genetic alteration may help identify biomarkers of the response and personalize the treatment for intermediate-stage HCC [115]. TACE is being investigated in combination with more than one ICI. The latest phase III EMERALD-1 study (NCT03778957) showed a statistically significant improvement in PFS using a combination of durvalumab plus bevacizumab plus TACE versus TACE in patients with unresectable HCC (15.0 vs. 8.2 months; HR 0.77; 95% CI 0.61–0.98; *p* = 0.032) [116]. The ongoing phase III EMERALD-3 trail (NCT05301842) is going to assess the combination of TACE plus durvalumab, tremelimumab and lenvatinib in patients with locoregional HCC not amenable to curative therapy.

The safety and effectiveness of TARE plus atezolizumab plus bevacizumab in patients with intermediate and advanced unresectable HCC were evaluated in the retrospective study. The promising results in this small patient cohort need further evaluation with a larger sample size [117]. The combination of TARE and ICI (durvalumab plus bevacizumab) is currently being investigated in the phase II EMERALD-Y90 study (NCT06040099) in patients with unresectable HCC eligible for embolization [118]. The phase II SOLID trial (NCT04124991) will assess the efficacy of TARE plus durvalumab in patients with locally advanced HCC. Moreover, the combination of SBRT and atezolizumab plus bevacizumab is being explored in phase I clinical trial NCT04857684 in patients with resectable HCC and the phase II NCT05137899 in HCC patients with the presence of portal vein tumour thrombus (PVTT) [119,120]. The phase Ib Notable-HCC trial (NCT05185531) is evaluating the effectiveness of tislelizumab and SBRT in BCLC stage 0–A HCC patients [121] (Table 5).

## 6. Role of Aetiology of the HCC in Survival Outcomes

Over the years, advances in HCC treatment have significantly improved patient outcomes. However, over time, it has been noticed that the response to treatment differs between patients with HCC of different aetiology, which may be due to molecular and immunohistochemical backgrounds. Lenvatinib showed greater efficacy in the treatment of NASH-related HCC compared to no NASH-related HCC, both in OS (22.2 vs. 15.1 months; *p* = 0.0006) and mPFS (7.5 vs. 6.5 months; *p* = 0.0436). Due to the fact that the incidence of lifestyle diseases, such as dyslipidemia, hypertension, obesity, or type II diabetes, is constantly increasing, the risk of NAFLD incidence increases, which translates into an increase in NASH-related HCC [122].

The meta-analysis of three major phase III studies (CheckMate-459, IMbrave150, and KEYNOTE-240) demonstrated that immunotherapy has better efficacy in OS in patients with HBV- or HCV-related HCC than for those with NASH-related HCC. This may be related to the phenotype of T lymphocytes in the tumour area, as noted in Pfister et al.’s study (2021). Anti-PD1 immunotherapy, instead of activating CD8 + PD1 + T cells, increased their exhaustion and created conditions for tumour growth through exacerbating liver inflammation. Supporting this, further analysis of NASH-HCC patients treated with PD-1/PD-L1 inhibitors showed reduced OS, further suggesting a different tumour immune environments [123]. Shalapour et al. pointed out that chronic inflammation in NAFLD is related to the accumulation of liver-resident immunoglobulin-A-producing (IgA+) cells with PD-L1 on their surface. The possibility of releasing IL-10 contributes to the suppression of CD8+ lymphocyte activity against tumour cells [124]. Moreover, microbiota specific to NAFLD-HCC may promote the expansion of Tregs, reduce CD8+ T cell activation, and increase the production of short-chain fatty acids (SCFAs) corelated with immune suppression. SCFAs may serve as a biomarker and help identify patients at higher risk of HCC progression [125]. The latest phase III trials showed that the survival benefit from ICIs was associated with HBV patients and NASH patients in HIMALAYA study and HBV patients in the CARES-310 study. Research is still ongoing to determine the precise influence of the microbiome on the response to ICIs in different HCC aetiologies [126].

## 7. Conclusions

HCC contributes to enormous mortality worldwide. Despite the continuous improvements in prevention and screening tests, the incidence of disease is constantly increasing. In many patients, HCC is diagnosed at an advanced stage; therefore, more effective systemic therapies need to be developed. Over the last two decades, there have been enormous advances in the treatment of HCC. TACE, SBRT, MWA, and RFA are constantly being improved in the treatment of low-advanced HCC. In addition to sorafenib, there are now lenvatinib and atezolizumab-bevacizumab as an option in the first-line treatment of advanced HCC. In the second-line treatment, cabozantinib or regorafenib is used. Huge progress has also been achieved in immunotherapy. Checkpoint inhibitors such as pembrolizumab or nivolumab may be used as second-line therapies in patients who cannot tolerate TKIs. Moreover, research has been ongoing for several years on methods combining ICI or TKI with TACE, SIRT, or SBRT to maximize the efficiency of treatment. Finally, for the first-line treatment of intermediate-stage HCC with criteria up to seven, the phase III REPLACE research will assess the safety and effectiveness of locoregional therapy with TACE or TARE versus regorafenib plus pembrolizumab. Advances in DNA or RNA sequencing technologies may also provide the latest information about the basis of HCC development, and molecular opportunities such as CRISPR-Cas9 screening may help find new signalling pathways, the inhibition of which would enable the inhibition of cancer cell proliferation.

## Figures and Tables

**Figure 1 cancers-16-04059-f001:**
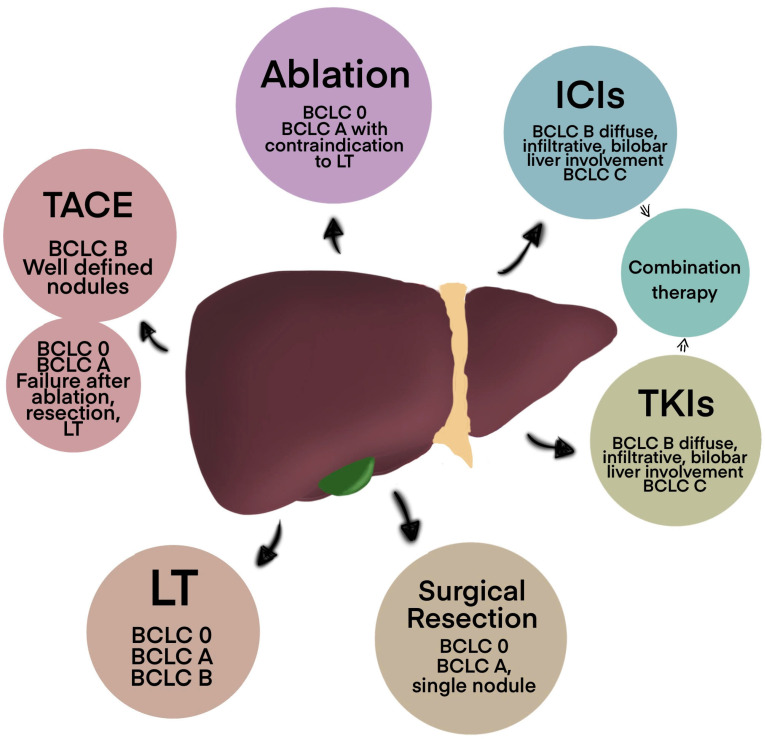
Current treatment methods in hepatocellular carcinoma. Abbreviations: LT—Liver Transplantation, TACE—Transarterial Chemoembolization, TKIs—Tyrosine Kinase Inhibitors, ICIs—Immune Checkpoint Inhibitors, BCLC—Barcelona Clinic Liver Cancer staging system.

**Table 1 cancers-16-04059-t001:** Barcelona Clinic Liver Cancer staging system with minimally invasive procedures distinction.

Stage		Performance Status (PS)	Liver Function (Child-Pugh Score)	Tumour Extend	
Stage 0	Very early stage	PS 0	Child-Pugh A	solitary lesion ≤ 2 cm in diameter	minimally invasive procedures
Stage A	Early stage	PS 0–2	Child-Pugh A–C	solitary lesion > 2 cm or early multifocal disease—up to 3 lesions measuring less than 3 cm	
Stage B	Intermediate stage	PS 0	Child-Pugh A–C	multifocal disease: >1 lesion with at least one > 3 cm, or >3 lesions regardless of their size	
Stage C	Advanced stage	PS 1–2	Child-Pugh A–C	vascular invasion and/or nodal disease and/or metastatic disease	
Stage D	End-stage disease	PS > 2	Child-Pugh C	any tumour burden	

**Table 2 cancers-16-04059-t002:** The primary and secondary outcome measures from phase III SHARP and REFLECT trials.

Trial	Design	*n*	HCC	mOS	TTP	PFS	ORR	AEs
SHARP NCT00105443	Sorafenib vs. placebo	602	Advanced HCC with Child-Pugh A with no prior systemic therapy	10.7 vs. 7.9 months	Median time to radiological progression 5.5 vs. 2.8 months Median time to symptomatic progression 4.1 vs. 4.9 months	-	-	Diarrhoea, fatigue, weight loss, anorexia, nausea, hand-foot skin reaction
				HR 0.69; 95% CI 0.55–0.87; *p* = 0.00583	HR 0.58; 95% CI 0.45–0.74; *p* < 0.001			
REFLECT NCT01761266	Lenvatinib vs. sorafenib	954	Unresectable HCC with no prior systemic therapy	13.6 vs. 12.3 months	8.9 vs. 3.7 months	7.4 vs. 3.7 months	24.1% vs. 9.2%	Hypertension, hand-foot skin reaction, dysphonia, weight loss, increased ALAT and ASPAT level, fatigue, diarrhoea
				HR 0.92; 95% CI 0.79–1.06				

Abbreviations: mOS—median Overall Survival, AE—Adverse Event, PFS—Progression-Free Survival, ORR—Objective Response Rate, TPP—Time to Progression, HCC—Hepatocellular Carcinoma, ALAT—Alanine Aminotransferase, ASPAT—Aspartate Aminotransferase, HR—Hazard Ratio.

**Table 3 cancers-16-04059-t003:** Clinical trials with ICIs.

Drug	Trial Name	Phase	Design	HCC	Endpoint	*n*	NCT
Nivolumab	CheckMate040	I	Nivolumab + ipilimumab vs. sorafenib	Advanced HCC	AEs, SAEs	659	NCT01658878
		II					
Nivolumab	CheckMate 459	III	Nivolumab vs. sorafenib	Advanced HCC	OS	743	NCT02576509
Atezolizumab	IMbrave150	III	Atezolizumab + bevacizumab vs. sorafenib	locally advanced or metastatic HCC, no prior systemic treatment	OS	558	NCT03434379
					PFS-IRF Per RECIST		
Pembrolizumab	MK-3475-224/KEYNOTE-224	II	Pembrolizumab	Cohort I: advanced HCC, no curative option after progression on sorafenib / intolerance of sorafenib	ORR	156	NCT02702414
				Cohort II: advanced HCC, not received treatment for systemic disease			
Pembrolizumab	MK-3475-394/KEYNOTE-394	III	Pembrolizumab + BSC vs. placebo + BSC	advanced HCC, previously systemically treated	OS	453	NCT03062358
Pembrolizumab	MK-3475-937/KEYNOTE-937	III	Pembrolizumab vs. placebo	Adjuvant therapy for HCC, complete radiological response after surgical resection or local ablation	RFS	950	NCT03867084
					OS		
Tremelimumab	-	II	durvalumab vs. tremelimumab monotherapyvs. durvalumab + tremelimumab vs. durvalumab + bevacizumab	Advanced HCC	DLTs	433	NCT02519348
					TEAEs		
					TESAEs		
Tremelimumab	HIMALAYA	III	durvalumab + tremelimumab vs. durvalumab monotherapy vs. sorafenib	unresectable HCC, no prior systemic therapy	OS	1324	NCT03298451
Tislelizumab	RATIONALE-208	II	Tislelizumab	Patients with previously treated HCC	OS	249	NCT03419897
Tislelizumab	RATIONALE-301	III	Tislelizumab vs. sorafenib	Unresectable HCC	OS	674	NCT03412773
Nivolumab	-	I	galunisertib + nivolumab	recurrent or refractory NSCLC or HCC	MTD	41	NCT02423343
		II					
Nivolumab	CheckMate 9DX	III	Nivolumab vs. placebo	HCC before complete resection or complete response after local ablation, high risk of recurrence	RFS	545	NCT03383458
Pembrolizumab	MK-7902-002/E7080-G000-311/LEAP-002	III	Lenvatinib + pembrolizumab vs. lenvatinib + placebo	first-line therapy for advanced HCC	PFS	794	NCT03713593
					OS		
Camrelizumab	CARES-310	III	Camrelizumab (SHR-1210) + apatinib vs. sorafenib	first-line therapy for advanced HCC	PFS	543	NCT03764293
					OS		
Sintilimab	ORIENT-32	II	Sintilimab + IBI305 vs. sorafenib	first-line therapy for advanced HCC	PFS	595	NCT03794440
		III			OS		
Camrelizumab	-	III	Camrelizumab + FOLFOX4 vs. placebo + FOLFOX4	Advanced HCC, no prior systemic treatment	OS	396	NCT03605706

Abbreviations: ICIs—Immune Checkpoint Inhibitors, RFS—Recurrence-Free Survival, OS—Overall Survival, AE—Adverse Event, SAEs—Serious Adverse Events, PFS—Progression-Free Survival, ORR—Objective Response Rate, PFS-IRF—Progression-Free Survival by Independent Review Facility-Assessment, RECIST—Response Evaluation Criteria in Solid Tumours, DLTs—Dose-Limiting Toxicities, TEAEs—Treatment-Emergent Adverse Events, TESAEs—Treatment-Emergent Serious Adverse Events, MTD—Maximum Tolerated Dose, NSCLC—Non-Small Cell Lung Cancer, HCC—Hepatocellular Carcinoma.

**Table 4 cancers-16-04059-t004:** Primary and secondary outcome measures in clinical trials with ICIs: CheckMate 459, IMbrave150, KEYNOTE-394, RATIONALE-301, HIMALAYA, LEAP—002, CARES-310, COSMIC-312.

Trial	Design	OS	ORR	PFS	TPP	DOR
CheckMate 459 NCT02576509	Nivolumab vs. sorafenib	16.4 vs. 14.7 months	15.4% vs. 7.0%	3.68 vs. 3.75 months	-	-
IMbrave150 NCT03434379	Atezolizumab + bevacizumab vs. sorafenib	CCOD—30 months	27.3% vs. 11.9%	6.83 vs. 4.27 months	8.57 vs. 5.59 months	NA vs. 6.28 months
		19.22 vs. 13.40 months				
KEYNOTE-394 NCT03062358	Pembrolizumab + BSC vs. placebo + BSC	14.6 vs. 13.0 months	12.7% vs. 1.3%	2.6 vs. 2.3 months	2.7 vs. 1.7 months	23.9 vs. 5.6 months
RATIONALE-301 NCT03412773	Tislelizumab vs. sorafenib	15.9 vs. 14.1 months	14.3% vs. 5.4%	2.1 vs. 3.4 months	-	36.1 vs. 11.0 months
HIMALAYA NCT03298451	Durvalumab 1500 mg + tremelimumab 300 mg × 1 dose vs. durvalumab 1500 mg + tremelimumab 75 mg × 4 doses vs. durvalumab 1500 mg monotherapy vs. sorafenib 400 mg × 2	16.43 vs. 16.36 vs. 16.56 vs. 13.77 months	20.1% vs. 17.0% vs. 17.0% vs. 5.1%	3.78 vs. 3.65 vs. 3.65 vs. 4.07 months	3.75 vs. 5.42 vs. 3.75 vs. 5.55 months	22.34 vs. 14.75 vs. 16.82 vs. 18.43 months
LEAP—002 NCT03713593	Lenvatinib + pembrolizumab vs. lenvatinib + placebo	21.2 vs. 19.0 months	26.1% vs. 17.5%	8.2 vs. 8.1 months	8.3 vs. 8.2 months	4.1 vs. 4.0 months
CARES-310 NCT03764293	Camrelizumab + apatinib vs. sorafenib	22.1 vs. 15.2 months	25% vs. 6%	5.6 vs. 3.7 months	-	14.8 vs. 9.2 months
COSMIC-312 NCT03755791	Cabozantinib + atezolizumab vs. sorafenib	16.5 vs. 15.5 months	-	6.9 vs. 4.3 months	-	-

Abbreviations: ICIs—Immune Checkpoint Inhibitors, OS—Overall Survival, AE—Adverse Event, SAEs—Serious Adverse Events, PFS—Progression-Free Survival, ORR—Objective Response Rate, DOR—Duration Of Response, CCOD—Clinical Cut-Off Date.

**Table 5 cancers-16-04059-t005:** ICI in combination with TACE, SBRT, and TARE. The clinical trials.

Phase	Drugs	Procedure	Setting	NCT
II	Nivolumab	TACE	intermediate stage HCC	NCT04268888 (TACE-3)
III				
II	Nivolumab	TACE	Intermediate Stage HCC	NCT03572582 (IMMUTACE)
III	Nivolumab with or without ipilimumab	TACE	Intermediate Stage HCC	NCT04340193 (CheckMate 74W)
II	Lenvatinib and Pembrolizumab	TACE	Advanced HCC	NCT04246177 (LEAP-012)
II	Apatinib and Camrelizumab	TACE	C staged HCC in BCLC classification	NCT04191889 (TRIPLET)
III	Lenvatinib and camrelizumab	TACE	BCLC C patients with the goal of conversion resection	NCT05738616 (LEN-TAC Study)
III	Atezolizumab and bevacizumab	TACE	Unresectable HCC	NCT047126430 (TALENTACE)
IIIb	Atezolizumab and bevacizumab	TACE	Intermediate stage HCC with no curative treatment option	NCT04803994 (ABC-HCC)
III	Durvalumab with or without bevacizumab	TACE	Unresectable HCC	NCT03778957 (EMERALD-1)
III	Durvalumab plus tremelimumab with or without bevacizumab	TACE	Locoregional HCC not amenable to curative therapy	NCT05301842 (EMERALD-3)
Early Phase I	Atezolizumab and bevacizumab	SBRT	Resectable HCC	NCT04857684
II	Atezolizumab and bevacizumab	SBRT	Solitary HCC with the presence of PVTT	NCT05137899 (ADVANCE HCC)
Ib	Tislelizumab	SBRT	Early-stage resectable HCC	NCT05185531 (Notable-HCC)
II	Durvalumab and bevacizumab	TARE	Unresectable HCC amenable to locoregional therapy	NCT06040099 (EMERALD-Y90)
II	Durvalumab	TARE	Locally advanced HCC	NCT04124991 (SOLID)

Abbreviations: ICI—Immune Checkpoint Inhibitor, HCC—Hepatocellular Carcinoma, PVTT—Portal Vein Tumour Thrombosis, TACE—Transarterial Chemoembolization, SBRT—Stereotactic Body Radiation Therapy, TARE—Transarterial Radioembolization.

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
