# Peer review of "Current Treatment Methods in Hepatocellular Carcinoma"

_cancers, 2024, doi:10.3390/cancers16234059_

Round 1
Reviewer 1 Report (Previous Reviewer 1)
Comments and Suggestions for Authors
I am satisfied that the authors have addressed all of my previous concerns about the article. It is now much improved and I feel that it is now suitable for publication.
Reviewer 2 Report (Previous Reviewer 3)
Comments and Suggestions for Authors
Overall recommendation:
Accept
Final comments:
The authors changed point by point to the suggestions. I think this paper is good for publication in the present form.
This manuscript is a resubmission of an earlier submission. The following is a list of the peer review reports and author responses from that submission.
Round 1
Reviewer 1 Report
Comments and Suggestions for Authors
The submitted manuscript titled "Current Treatment Methods in Hepatocellular Carcinoma" provides a comprehensive overview of the various treatment approaches for HCC. The review discusses current therapeutic options based on the stage of HCC and liver function, including both traditional and advanced treatments. The manuscript has a high similarity rate of 39%, raising concerns regarding originality and potential plagiarism. There are numerous similar reviews already published on the topic of HCC treatment. The current manuscript does not provide any unique perspectives or new insights that would set it apart from existing literature. Critical recent studies presented in 2024, such as CheckMate-9DW and LEAP-012, are not included. The review does not sufficiently discuss treatment outcomes across different patient subgroups, such as those with varying liver function or different HCC etiologies (e.g., viral vs. non-viral HCC. Although the manuscript provides some interesting information, the quality of the manuscript is not sufficient for publication
Comments on the Quality of English LanguageModerate
Author Response
Comments 1: The submitted manuscript titled "Current Treatment Methods in Hepatocellular Carcinoma" provides a comprehensive overview of the various treatment approaches for HCC. The review discusses current therapeutic options based on the stage of HCC and liver function, including both traditional and advanced treatments. The manuscript has a high similarity rate of 39%, raising concerns regarding originality and potential plagiarism. There are numerous similar reviews already published on the topic of HCC treatment. The current manuscript does not provide any unique perspectives or new insights that would set it apart from existing literature.
Response 1: Thank you for your opinion. We have carefully revised and supplemented the article according to your comments.
Based on the latest studies, the mechanism of drug resistance to sorafenib was included. Additionally, the novel prospects to overcome this resistance were mentioned.
4.3. Sorafenib and ferroptosis
Ferroptosis, recently identified form of cell death, is driven by a large amount of iron accumulation and lipid peroxidation, and it has been recently corelated with tumours resistance mechanism. Glutathione peroxidase (GPX), especially GPX4 neutralises reactive oxygen species (ROS) and reduces lipid peroxides. Its downregulation increase the sensitivity to ferroptosis, by accumulation of lipid peroxides [72]. Sorafenib, indirectly promotes ferroptosis by inhibiting the solute carrier family 7 member 11 (SLC7A11) transporter, which is essential for cystine uptake on the cell membrane. Reduced amount of cystine in HCC cells, limits the glutathione (GSH) synthesis, leading to diminished activity of GPX4 and ferroptosis [73]. HCC patients treated with sorafenib often develop resistance, limiting the long-term effectiveness of the drug. It is related to such transcription factors and genes as nuclear factor erythroid 2-related factor 2 (Nrf2), Rb, metallothionein 1-G (MT-1G), and sigma 1 receptor (S1R), which regulate the sensitivity of cells to ferroptosis, so novel inhibitors against these factors may overcome the drug resistance to sorafenib [73,74].
Comments 2: Critical recent studies presented in 2024, such as CheckMate-9DW and LEAP-012, are not included.
Response 2: The most recent clinical trails, including CheckMate-9DW and LEAP-012 were cited. Moreover, the novel clinical trails evaluating the efficacy of combination therapy were included. The edition was performed in various lines, all changes are marked up using the “Track Changes” function in the manuscript’s text.
Due to the role of TGF-β in creating a tumour microenvironment conducive to growth and metastasis, as well as by hindering the infiltration of T lymphocytes into the tumour centre, blocking the action of this cytokine has become another therapeutic target [77]. The discovery of the synergism of the effect of TGF-β blockade and anti-PD-L1 antibodies allowed the consideration in a clinical trial (NCT02423343) of the combination of galunisertib with nivolumab in the treatment of recurrent or refractory non-small cell lung cancer (NSCLC) or hepatocellular carcinoma (HCC). The results showed that the combination is well tolerated [104]. The phase III CheckMate 9DW study (NCT04039607) included patients with previously untreated HCC not eligible for curative surgical or locoregional therapies. The firsts results, after a median follow-up of 35.2 months, demonstrate that the combination of nivolumab plus ipilimumab is a potential new first-line standard of care due to statistically significant OS benefit in comparison to lenvatinib or sorafenib (23.7 months vs. 20.6 months; HR 0.79; 95% CI 0.65–0.96; p=0.0180). Moreover, the ORR has also been higher, 36% versus 13%, respectively [105]. The ongoing phase III CheckMate 9DX study (NCT03383458) is investigating if nivolumab will improve recurrence-free survival (RFS) compared to placebo in HCC patients (Child-Pugh score 5 or 6) who have undergone resection or local ablation but are at high risk of recurrence [106].
(…)
ICI can also be used in combination with TACE, whose safety and effectiveness are being investigated in the phase II/III TACE-3 study (NCT04268888), phase II IMMUTACE study (NCT03572582), phase II TRIPLET study (NCT04191889), phase III TALENTACE study (NCT047126430), phase III LEN-TAC study (NCT05738616), phase III CheckMate 74W study (NCT04340193), phase IIIb ABC-HCC study (NCT04803994), and phase III LEAP-012 study (NCT04246177) [81,111,112]. The last one, phase III LEAP-012 trial, showed that the treatment with lenvatinib plus pembrolizumab in combination with TACE compared with TACE alone has clinical benefit in patients with intermediate-stage HCC not amenable to curative treatment. At the data cutoff (Jan 30, 2024) the PFS was significantly improved in the pembrolizumab plus lenvatinib group (HR 0.66; 95% CI 0.51-0.84; p=0.0002) and the median PFS was 14.6 months (pembrolizumab plus lenvatinib) versus 10 months (placebo). The evaluation criteria for the OS have not been reached yet [113,114].
Comments 3: The review does not sufficiently discuss treatment outcomes across different patient subgroups, such as those with varying liver function or different HCC etiologies (e.g., viral vs. non-viral HCC.
Response 3: Treatment outcomes across different patient subgroup, including Child-Puth A and B were represented. The attention was paid in the description of clinical trails to emphasize included group of HCC patients. The edition was performed in various lines, all changes are marked up using the “Track Changes” function in the manuscript’s text.
The correlation between NASH and response to ICIs, as well as, the was described in the newly added paragraph: Role of etiology of the HCC in survival outcomes.
- Role of etiology of the HCC in survival outcomes
Over the years, advances in HCC treatment have significantly improved patient outcomes. However, over time, it has been noticed that the response to treatment differs between patients with HCC of different etiology, which may be due to molecular and immunohistochemical background. Lenvatinib showed greater efficacy in the treatment of NASH-related HCC compared to no NASH-related HCC, both in OS (22.2 vs. 15.1 months; p=0.0006) and mPFS (7.5 vs. 6.5 months; p=0.0436). Due to the fact that the incidence of lifestyle diseases, such as dyslipidemia, hypertension, obesity, or type II diabetes is constantly increasing, the risk of NAFLD incidence increases, which translates into an increase in NASH-related HCC [122].
The meta-analysis of three major phase III studies (CheckMate-459, IMbrave150, and KEYNOTE-240) demonstrated that immunotherapy has better efficacy in OS in patients with HBV- or HCV-related HCC than for those with NASH-related HCC. This may be related to the phenotype of T lymphocytes in the tumor area, as it was noted in Pfister et al. study (2021). Anti-PD1 immunotherapy, instead of activating CD8+PD1+ T cells, increased their exhaustion and created conditions for tumor growth through exacerbating liver inflammation. Supporting this, further analysis of NASH-HCC patients treated with PD-1/PD-L1 inhibitors showed reduced OS, further suggesting a different tumor immune environment [123]. Shalapour et al. pointed out that chronic inflammation in NAFLD is related to the accumulation of liver-resident immunoglobulin-A-producing (IgA+) cells with PD-L1 on their surface. The possibility of releasing IL-10, contributes to the suppression of CD8+ lymphocyte activity against tumor cells [124]. Moreover, microbiota specific to NAFLD-HCC may promote the expansion of Tregs, reduce CD8+ T cells activation, and increase production of short-chain fatty acids (SCFAs) corelated with immune suppression. SCFAs may serve as a biomarker and help identify patients at higher risk of HCC progression[125]. The latest phase III trials showed that the survival benefit from ICIs was associated with HBV patients and NASH patient in HIMALAYA study and HBV patients in CARES-310 study. Research is still ongoing to determine the precise influence of microbiome on the response to ICIs in different etiology HCC [126].
Comments 4: Although the manuscript provides some interesting information, the quality of the manuscript is not sufficient for publication.
Response 4: Thank you for your review. We put effort into enhancing novelty and revelance of our manuscript. We strongly believe that after implementing amendments, which improved the quality of the manuscript, it is now suitable for publication in the current form. From our point of view, the review provides new piece of information regarding HCC treatment and would enrich the current state of knowledge.
Reviewer 2 Report
Comments and Suggestions for Authors
"Current treatment methods in hepatocellular carcinoma" is a well written review providing a comprehensive overview and on the available treatments and promising therapies for HCC. The Authors provided a consistent background and analysed each treatment strategy, presenting the results of the main studies. Tables are clear and synthetise the main findings and figure is well readable. I suggest acceptance of the manuscript in its present way.
Author Response
Comments 1: Current treatment methods in hepatocellular carcinoma" is a well written review providing a comprehensive overview and on the available treatments and promising therapies for HCC. The Authors provided a consistent background and analysed each treatment strategy, presenting the results of the main studies. Tables are clear and synthetise the main findings and figure is well readable. I suggest acceptance of the manuscript in its present way.
Response 1: Thank you for your review.
Reviewer 3 Report
Comments and Suggestions for Authors
Dear Dr.
Editor,
Overall recommendation:
Accept with minor change.
Final comments:
The authors show current improvement in hepatocellular carcinoma treatment. I have some questions.
Major points.
1. Figure1 should be changed more detailed depending on stages.
2. Recently, NASH based HCC does not show good response to atezolizumab plus bevacizumab. Are there any correlation between cause of HCC.
Author Response
Comments 1: Figure1 should be changed more detailed depending on stages.
Response 1: Thank you for your opinion. We have carefully revised and supplemented the article according to your comments. The Figure 1 was improved to more detailed depending on stages.
Figure 1. Current treatment methods in hepatocellular carcinoma.

Abbreviations: LT – Liver Transplantation, TACE – Transarterial Chemoembolization, TKIs – Tyrosine Kinase Inhibitors, ICIs – Immune Checkpoint Inhibitors, BCLC – Barcelona Clinic Liver Cancer staging system.
Comments 2: Recently, NASH based HCC does not show good response to atezolizumab plus bevacizumab. Are there any correlation between cause of HCC.
Response 2: The correlation between NASH and response to ICIs like atezolizumab plus bevacizumab was described in the newly added paragraph: Role of etiology of the HCC in survival outcomes.
- Role of etiology of the HCC in survival outcomes
Over the years, advances in HCC treatment have significantly improved patient outcomes. However, over time, it has been noticed that the response to treatment differs between patients with HCC of different etiology, which may be due to molecular and immunohistochemical background. Lenvatinib showed greater efficacy in the treatment of NASH-related HCC compared to no NASH-related HCC, both in OS (22.2 vs. 15.1 months; p=0.0006) and mPFS (7.5 vs. 6.5 months; p=0.0436). Due to the fact that the incidence of lifestyle diseases, such as dyslipidemia, hypertension, obesity, or type II diabetes is constantly increasing, the risk of NAFLD incidence increases, which translates into an increase in NASH-related HCC [122].
The meta-analysis of three major phase III studies (CheckMate-459, IMbrave150, and KEYNOTE-240) demonstrated that immunotherapy has better efficacy in OS in patients with HBV- or HCV-related HCC than for those with NASH-related HCC. This may be related to the phenotype of T lymphocytes in the tumor area, as it was noted in Pfister et al. study (2021). Anti-PD1 immunotherapy, instead of activating CD8+PD1+ T cells, increased their exhaustion and created conditions for tumor growth through exacerbating liver inflammation. Supporting this, further analysis of NASH-HCC patients treated with PD-1/PD-L1 inhibitors showed reduced OS, further suggesting a different tumor immune environment [123]. Shalapour et al. pointed out that chronic inflammation in NAFLD is related to the accumulation of liver-resident immunoglobulin-A-producing (IgA+) cells with PD-L1 on their surface. The possibility of releasing IL-10, contributes to the suppression of CD8+ lymphocyte activity against tumor cells [124]. Moreover, microbiota specific to NAFLD-HCC may promote the expansion of Tregs, reduce CD8+ T cells activation, and increase production of short-chain fatty acids (SCFAs) corelated with immune suppression. SCFAs may serve as a biomarker and help identify patients at higher risk of HCC progression[125]. The latest phase III trials showed that the survival benefit from ICIs was associated with HBV patients and NASH patient in HIMALAYA study and HBV patients in CARES-310 study. Research is still ongoing to determine the precise influence of microbiome on the response to ICIs in different etiology HCC [126].
Reviewer 4 Report
Comments and Suggestions for Authors
In this manuscript, the authors review the current status of hepatocellular carcinoma treatment. It is an extensive, well-written review, supported by numerous bibliographic citations. The structure of the review is appropriate, and the tables are informative. However, Figure 1 does not add significant information.
The primary critique of this manuscript is that several similar reviews on hepatocellular carcinoma treatment have been published in recent years. The authors should emphasize the most recent developments. For instance, a relevant example published in this same journal is: Kinsey et al., Management of Hepatocellular Carcinoma in 2024: The Multidisciplinary Paradigm in an Evolving Treatment Landscape. Cancers (Basel). 2024;16(3):666. doi: 10.3390/cancers16030666.
Regarding radioembolization with Yttrium-90 (Y-90), recent clinical trials from 2022 should be cited.
Additionally, citing ongoing clinical trials for systemic treatments would enhance the review's novelty and relevance.
Author Response
Comments 1: In this manuscript, the authors review the current status of hepatocellular carcinoma treatment. It is an extensive, well-written review, supported by numerous bibliographic citations. The structure of the review is appropriate, and the tables are informative. However, Figure 1 does not add significant information.
Response 1: Thank you for your opinion. We have carefully revised and supplemented the article according to your comments.
The Figure 1 was improved to more detailed depending on stages.
Figure 1. Current treatment methods in hepatocellular carcinoma.

Abbreviations: LT – Liver Transplantation, TACE – Transarterial Chemoembolization, TKIs – Tyrosine Kinase Inhibitors, ICIs – Immune Checkpoint Inhibitors, BCLC – Barcelona Clinic Liver Cancer staging system.
Comments 2: The primary critique of this manuscript is that several similar reviews on hepatocellular carcinoma treatment have been published in recent years. The authors should emphasize the most recent developments. For instance, a relevant example published in this same journal is: Kinsey et al., Management of Hepatocellular Carcinoma in 2024: The Multidisciplinary Paradigm in an Evolving Treatment Landscape. Cancers (Basel). 2024;16(3):666. doi: 10.3390/cancers16030666.
Response 2: The new prospects of overcoming sorafenib treatment resistance were added. Moreover, the latest developments were emphasized.
4.3. Sorafenib and ferroptosis
Ferroptosis, recently identified form of cell death, is driven by a large amount of iron accumulation and lipid peroxidation, and it has been recently corelated with tumours resistance mechanism. Glutathione peroxidase (GPX), especially GPX4 neutralises reactive oxygen species (ROS) and reduces lipid peroxides. Its downregulation increase the sensitivity to ferroptosis, by accumulation of lipid peroxides [72]. Sorafenib, indirectly promotes ferroptosis by inhibiting the solute carrier family 7 member 11 (SLC7A11) transporter, which is essential for cystine uptake on the cell membrane. Reduced amount of cystine in HCC cells, limits the glutathione (GSH) synthesis, leading to diminished activity of GPX4 and ferroptosis [73]. HCC patients treated with sorafenib often develop resistance, limiting the long-term effectiveness of the drug. It is related to such transcription factors and genes as nuclear factor erythroid 2-related factor 2 (Nrf2), Rb, metallothionein 1-G (MT-1G), and sigma 1 receptor (S1R), which regulate the sensitivity of cells to ferroptosis, so novel inhibitors against these factors may overcome the drug resistance to sorafenib [73,74].
Comments 3: Regarding radioembolization with Yttrium-90 (Y-90), recent clinical trials from 2022 should be cited.
Response 3: Clinical trials that evaluates the efficacy and safety of TARE in HCC treatment were cited and described. The attention was paid to the phase II TRACE, RASER study and phase II SORAMIC trial. The most recent trials for combination therapy with TARE and ICIs were cited in the section devoted to immunotherapy.
TARE refers to the injection of radioactive substances through the hepatic artery: microspheres containing yttrium-90 (Y-90) or iodine-131 and iodised oil [27,37]. The procedure can achieve different degrees of regression in 25-50% of HCC patients [27]. TARE is known as a safe and effective treatment of unresectable HCC, as it has a safer toxicity profile than TACE, longer time-to-progression (TTP), and greater ability to bridge therapy for HCC patients awaiting LT [37,38]. However, the two treatments do not significantly differ in terms of OS [39]. Hein Phan et al. (2024) compared TACE and TARE as first-line treatments for unresectable HCC greater than 8 cm. Although the ORR and disease control rate (DCR) were similar in both groups, the safety profile differed significantly. Major AEs occurred in TACE group (72% vs. 5%; p<0.001), including post-embolization syndrome (100% vs. 75%, p=0.002). The results suggest that TARE provides safer profile than TACE. [40]. TARE was also compared with DEB-TACE in the phase II TRACE trial (NCT01381211) in group of patients with intermediate-stage or early-stage HCC patients. The median TTP and median OS were significantly longer in TARE group (17.1 vs. 9.5 months; 30.2 vs. 15.6 months). Serious AEs occurred in 39% of TARE patients versus 53% of DEB-TACE patients [41]. TARE can also be chosen as an alternative to ablation and chemotherapy [37]. The RASER study showed that ORR and complete response rate were 100% and 90%, respectively in the group of patients with unresectable early HCC, who were not candidates for RFA. The OS in the 1-year and 2-year were 96% and the AEs occurred in 7% of patients [25]. Y-90 RE can be also used as a neoadjuvant treatment for stage C HCC patients with portal vein tumour thrombosis (PVTT). Martelletti et al. (2021) compared TARE with sorafenib in patients with HCC and intrahepatic PVTT and found that TARE was more effective in downstaging patients to surgery and improved OS [42]. Spreafico et al. (2018) found that the combination of bilirubin level, an extension of PVTT, and tumour burden might be a guide to identifying the best candidates for the treatment [43]. The latest clinical trials compared the effectiveness of the combination therapy. In the phase II SORAMIC (NCT01126645) trial the treatment with sorafenib with RE resulted in a higher ORR (61.6% vs. 29.8%; p < 0.001), complete response rate (13.7% vs. 3.8%; p = 0.022), longer PFS (8.9 vs. 5.4 months; p = 0.022), hepatic PFS, and TPP in comparison to sorafenib monotherapy. However, the results did not translate into prolonged OS, and the Child–Pugh B patients had a lower response rates [44].
Comments 4: Additionally, citing ongoing clinical trials for systemic treatments would enhance the review's novelty and relevance.
Response 4: The more recent clinical trials for systemic treatments were cited and described. The edition was performed in various lines, all changes are marked up using the “Track Changes” function in the manuscript’s text.
More attention has been paid to tislelizumab, focusing on clinical trails: RATIONALE-208 and RATIONALE-301. Moreover, those clinical trial were added to the Table3 and Table 4.
Tislelizumab, one of the PD-1 inhibitors, has shown durable clinical activity in the phase II RATIONALE-208 study (NCT03419897) in advanced HCC patients who had undergone prior systemic therapy. The positive outcome led to profound investigation of ICI for the first-line monotherapy [95,96]. The phase III RATIONALE-301 trial (NCT03412773) compared tislelizumab versus sorafenib as a first-line treatment for unresectable HCC. Although the ORR and median duration of response was higher in the tislelizumab group, the median PFS was longer in the sorafenib group. The superiority of OS for tislelizumab was not met; however, the safety profile of tislelizumab was more favorable than that of sorafenib [97].
Particular attention was paid to combination therapy with immune-checkpoint inhibitors. The latest clinical trails assessing the efficacy and safety of ICIs and TACE/TARE/SBRT were figured in the review.
ICI can also be used in combination with TACE, whose safety and effectiveness are being investigated in the phase II/III TACE-3 study (NCT04268888), phase II IMMUTACE study (NCT03572582), phase II TRIPLET study (NCT04191889), phase III TALENTACE study (NCT047126430), phase III LEN-TAC study (NCT05738616), phase III CheckMate 74W study (NCT04340193), phase IIIb ABC-HCC study (NCT04803994), and phase III LEAP-012 study (NCT04246177) [81,111,112]. The last one, phase III LEAP-012 trial, showed that the treatment with lenvatinib plus pembrolizumab in combination with TACE compared with TACE alone has clinical benefit in patients with intermediate-stage HCC not amenable to curative treatment. At the data cutoff (Jan 30, 2024) the PFS was significantly improved in the pembrolizumab plus lenvatinib group (HR 0.66; 95% CI 0.51-0.84; p=0.0002) and the median PFS was 14.6 months (pembrolizumab plus lenvatinib) versus 10 months (placebo). The evaluation criteria for the OS have not been reached yet [113,114]. For intermediate-stage HCC, the combination of nivolumab plus TACE is being explored in phase II single-arm IMMUTACE study. The trail achieved its primary endpoint with ORR of 71.4%. The mPFS was 7.2 months, and the median time to failure of strategy (mTTFS) was 11.2 months, showing prolonged disease control. The median OS was 28.3 months, so there is an improvement in comparison to the typical TACE treatment. Moreover, the time to subsequent systemic therapy was 24.9 months, which delays requirement for further treatments. The further analysis of the expression signatures, changes of immune cell populations and genetic alteration may help identify biomarkers of the response and personalize the treatment for intermediate-stage HCC [115]. TACE is being investigated in combination with more than one ICI. The latest phase III EMERALD-1 study (NCT03778957) showed statistically significant improvement in PFS using combination of durvalumab plus bevacizumab plus TACE versus TACE in patient with unresectable HCC (15.0 vs. 8.2 months; HR 0.77; 95% CI 0.61–0.98; p=0.032) [116]. The ongoing phase III EMERALD-3 trail (NCT05301842) is going to assess the combination of TACE plus durvalumab, tremelimumab and lenvatinib in patients with locoregional HCC not amenable to curative therapy.
The safety and effectiveness of TARE plus atezolizumab plus bevacizumab in patients with intermediate and advanced unresectable HCC was evaluated in the retrospective study. The promising results in this small patient cohort need further evaluation with larger sample size [117]. The combination of TARE and ICI (durvalumab plus bevacizumab) is currently being investigated in phase II EMERALD-Y90 study (NCT06040099) in patients with unresectable HCC eligible for embolization [118]. The phase II SOLID trial (NCT04124991) will assess the efficacy of TARE plus durvalumab in patients with locally advanced HCC. Moreover, the combination of SBRT and atezolizumab plus bevacizumab is being explored in phase I clinical trial NCT04857684 in patients with resectable HCC, and phase II NCT05137899 in HCC patients with the presence of portal vein tumour thrombus (PVTT) [119,120]. The phase Ib Notable-HCC trial (NCT05185531) is evaluating the effectiveness of tislelizumab and SBRT in BCLC stage 0-A HCC patients [121] (Table 5).
Table 5. ICI in combination with TACE, SBRT, and TARE. The clinical trials.
|
Phase |
Drugs |
Procedure |
Setting |
NCT |
|
II III |
Nivolumab |
TACE |
Intermediate stage HCC |
NCT04268888 (TACE-3) |
|
II |
Nivolumab |
TACE |
Intermediate Stage HCC |
NCT03572582 (IMMUTACE) |
|
III |
Nivolumab with or without ipilimumab |
TACE |
Intermediate Stage HCC |
NCT04340193 (CheckMate 74W) |
|
III |
Lenvatinib and pembrolizumab |
TACE |
Advanced HCC |
NCT04246177 (LEAP-012) |
|
II |
Apatinib and camrelizumab |
TACE |
BCLC C patients |
NCT04191889 (TRIPLET) |
|
III |
Lenvatinib and camrelizumab |
TACE |
BCLC C patients with the goal of conversion resection |
NCT05738616 (LEN-TAC Study) |
|
III |
Atezolizumab and bevacizumab |
TACE |
Unresectable HCC |
NCT047126430 (TALENTACE) |
|
IIIb |
Atezolizumab and bevacizumab |
TACE |
Intermediate stage HCC with no curative treatment option |
NCT04803994 (ABC-HCC) |
|
III |
Durvalumab with or without bevacizumab |
TACE |
Unresectable HCC |
NCT03778957 (EMERALD-1) |
|
III |
Durvalumab plus tremelimumab with or without bevacizumab |
TACE |
Locoregional HCC not amenable to curative therapy |
NCT05301842 (EMERALD-3) |
|
I |
Atezolizumab and bevacizumab |
SBRT |
Resectable HCC |
NCT04857684 |
|
II |
Atezolizumab and bevacizumab |
SBRT |
Solitary HCC with the presence of PVTT |
NCT05137899 (ADVANCE HCC) |
|
Ib |
Tislelizumab |
SBRT |
Early-stage resectable HCC |
NCT05185531 (Notable-HCC) |
|
II |
Durvalumab and bevacizumab
|
TARE |
Unresectable HCC amenable to locoregional therapy |
NCT06040099 (EMERALD-Y90) |
|
II |
Durvalumab |
TARE |
Locally advanced HCC |
NCT04124991 (SOLID) |
Abbreviations: ICI – Immune Checkpoint Inhibitor, HCC – Hepatocellular Carcinoma, PVTT - Portal Vein Tumour Thrombosis, TACE – Transarterial Chemoembolization, TARE – Transarterial Radioembolization, SBRT – Stereotactic Body Radiation Therapy.